# Optimal Architectures for Judging LLM Outputs using LLMs

## Abstract

This paper explores optimal architectures for evaluating the outputs of large language models (LLMs) using LLMs themselves. We propose a novel framework that interprets LLMs as advocates within an ensemble of interacting agents, allowing them to defend their answers and reach conclusions through a judge and jury system. This approach offers a more dynamic and comprehensive evaluation process compared to traditional human-based assessments or automated metrics. We discuss the motivation behind this framework, its key components, and comparative advantages. We also present a probabilistic model to evaluate the error reduction achieved by iterative advocate systems. Finally, we outline experiments to validate the effectiveness of multi-advocate architectures and discuss future research directions.

## 1 Introduction

The rapid advancement of large language models (LLMs) has revolutionized the field of natural language processing, enabling the development of increasingly sophisticated AI systems capable of generating human-like text, engaging in dialogue, and performing complex language tasks (5). As these models grow in size and capability, the challenge of accurately evaluating their performance and aligning their outputs with human preferences has become increasingly critical (3; 15; 49).

Traditional evaluation methods, such as human assessments and automated metrics, often struggle to capture the nuances and complexities of LLM outputs, leading to a gap between model performance and user expectations (7; 17; 24). Human evaluations are time-consuming, expensive, and prone to inconsistency and bias (12; 27), while automated metrics frequently fail to align with human judgments, particularly in open-ended generation tasks (29; 13; 22).

To address these challenges, we propose a novel framework for evaluating LLM outputs using LLMs themselves as interacting agents in a courtroom-inspired, multi-agent system. Our approach draws inspiration from various fields, including decision theory, economics, psychology, legal theory, and voting theory, to develop a more dynamic, contextual, and comprehensive assessment process.

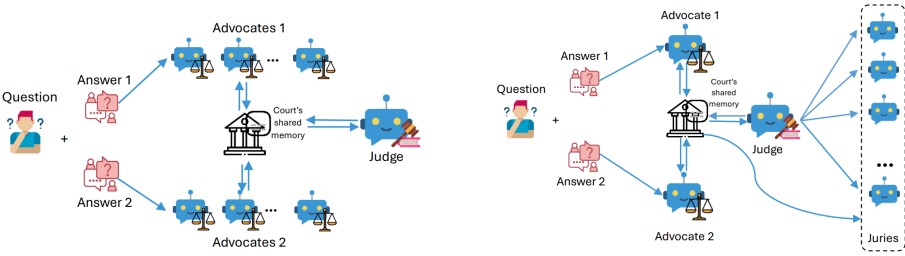

Figure 1: Illustrations of the architectures: the **MORE** architecture (left) employs multiple advocates per answer, while the **SAMRE** architecture (right) utilizes a single advocate per answer but allows for multiple rounds of evaluation.

## 1.1 MOTIVATION FROM DECISION THEORY AND LEGAL THEORY

Our approach is motivated by various approaches proposed in literature on designing systems with agents of varying capabilities and incentives. In what follows, we review a few motivating frameworks.

### DECISION THEORY AND BOUNDED RATIONALITY

Decision theory provides a foundation for understanding how agents make choices under uncertainty and constraints (44; 21). The concept of bounded rationality, introduced by Herbert A. Simon (39; 40), acknowledges that decision-makers often operate with limited information, cognitive resources, and time, leading to satisficing rather than optimizing behavior. In the context of LLM evaluation, our LLM advocates framework addresses bounded rationality by distributing the cognitive load across multiple specialized agents, each focusing on a specific aspect of the evaluation process. This division of labor allows for a more efficient and targeted assessment, mitigating the constraints faced by individual agents and enabling the system to converge on more accurate and informative evaluations.

### PSYCHOLOGICAL THEORIES OF PERSUASION AND ARGUMENTATION

Psychological theories of persuasion and argumentation, such as the Elaboration Likelihood Model (36) and the Heuristic-Systematic Model (8), provide valuable insights into the factors that influence the effectiveness of arguments and the formation of judgments. These theories highlight the importance of central and peripheral routes to persuasion, as well as the role of heuristics and biases in shaping perceptions and decisions.

Our LLM advocates framework incorporates elements of persuasion and argumentation theory by encouraging LLMs to present well-structured, compelling arguments that appeal to both central and peripheral routes of persuasion. By exposing the outputs to scrutiny from opposing advocates and subjecting them to the judgment of impartial LLM juries, our system helps to identify and mitigate the influence of heuristics and biases, leading to more robust and reliable evaluations.

### LEGAL THEORIES OF ADVERSARIAL PROCESS AND JURISPRUDENCE

Legal theories of adversarial process and jurisprudence emphasize the importance of structured debate, cross-examination, and impartial judgment in uncovering truth and reaching fair outcomes (46; 16; 41). The adversarial system, which lies at the heart of many legal traditions, relies on the clash of opposing arguments to test the strength of evidence and reasoning, while the role of neutral judges and juries ensures that decisions are based on a balanced consideration of the facts and arguments presented.

Our LLM advocates framework draws inspiration from the adversarial legal process, casting LLMs as advocates tasked with presenting and defending competing arguments, while other LLMs serve as impartial judges and juries. This structure promotes a thorough and rigorous examination of LLM outputs, exposing weaknesses and inconsistencies that may be overlooked by traditional evaluation methods. By emulating the checks and balances of the legal system, our framework aims to produce more accurate, unbiased, and trustworthy assessments of LLM performance.

Furthermore, we also draw inspiration from Voting and social choice theories, which study the design of collective decision-making systems, considering factors such as preference aggregation, strategic behavior, and fairness (2; 18; 38). Our LLM advocates framework incorporates recommendations of voting theory and social choice by employing multi-layer jury systems to aggregate the judgments of multiple LLM agents. By exploring different voting schemes (e.g., majority rule, Borda count, or pairwise comparisons), our framework can adapt to the specific requirements and constraints of different evaluation contexts.

## 1.2 NOVEL CONTRIBUTIONS AND PAPER STRUCTURE

Building on the insights from these diverse fields, our paper makes several novel contributions to the problem of LLM evaluation:

1. We propose a dynamic, multi-agent framework that casts LLMs as interacting advocates, judges, and juries, enabling a more comprehensive and contextual assessment of LLM outputs.

2. We introduce a courtroom-inspired architecture that leverages the power of structured debate, cross-examination, and impartial judgment to uncover strengths, weaknesses, and inconsistencies in LLM responses.

3. We draw on theories of bounded rationality, incentive design, persuasion, argumentation, and adversarial process to inform the design of our LLM advocates framework, ensuring that the system promotes accurate, unbiased, and trustworthy evaluations.

4. We explore the use of voting theory and social choice principles to design effective jury systems for aggregating LLM judgments, promoting fair and representative assessments while mitigating the influence of strategic behavior and individual biases.

The remainder of this paper is structured as follows: Section 2 reviews related work on LLM evaluation, highlighting the limitations of existing approaches and the need for more sophisticated assessment frameworks. Section 3 introduces our LLM advocates framework, detailing its courtroom-inspired architecture, the roles and interactions of the various LLM agents, and the underlying theoretical principles that inform its design. Section 4 presents a series of experiments and case studies demonstrating the effectiveness of our framework in evaluating LLM outputs across a range of tasks and domains. Section 5 discusses the implications of our findings, the limitations of our approach, and potential directions for future research. Finally, Section 6 concludes the paper, summarizing our contributions and outlining the broader impact of our work on the development of reliable, transparent, and accountable AI systems.

## 2 RELATED WORK

The evaluation of language models has been a longstanding challenge in the field of natural language processing, with the rapid growth of LLMs in recent years bringing this issue to the forefront. As these models have increased in size and capability, the need for robust, comprehensive, and theoretically grounded evaluation methods has become increasingly apparent. This section reviews relevant literature on LLM evaluation, drawing on insights from human-computer interaction, psychometrics, multi-agent systems, and social choice theory to highlight the limitations of existing approaches and the potential for more sophisticated assessment frameworks.

### 2.1 HUMAN-BASED EVALUATIONS AND THE CHALLENGE OF SUBJECTIVITY

Human judgments have long been considered the gold standard for evaluating the quality of language model outputs, with platforms like LMSYS Chatbot Arena (10) and others (47; 23) providing structured environments for collecting human ratings and preferences. However, the subjectivity and variability inherent in human evaluations pose significant challenges for the reliable and consistent assessment of LLMs (12; 27).

Research in human-computer interaction and cognitive psychology has shown that factors such as individual differences, task framing, and cognitive biases can significantly influence human judgments of AI systems (20; 14; 6). For example, the anchoring effect (42) and the halo effect (33) can lead to over- or under-estimation of LLM performance based on initial impressions or salient features, while the confirmation bias (32) can cause evaluators to seek out information that supports their preconceptions.

Moreover, the reliance on reinforcement learning from human feedback (RLHF) (4; 11; 50) for aligning LLMs with user expectations introduces additional challenges, as the pool of reinforcers may not be representative of the general user population (31). This can lead to models that are optimized for the preferences of a narrow subset of users, potentially exacerbating issues of bias, fairness, and accountability (30; 19).

### 2.2 AUTOMATED METRICS AND THE LIMITS OF REFERENCE-BASED EVALUATION

To address the scalability and consistency issues of human evaluations, researchers have developed various automated metrics for assessing LLM performance across different tasks, such as BLEU

(35) for machine translation, ROUGE (28) for summarization, and exact match (EM) and F1 scores (37) for question answering. These metrics provide a standardized and efficient means of evaluating LLMs, enabling the comparison of different models and the tracking of progress over time.

However, the reliance of these metrics on reference-based evaluation, where model outputs are compared against a fixed set of ground-truth answers, has been shown to have significant limitations (29; 13; 22). In open-ended generation tasks, such as dialogue and creative writing, there may be a wide range of acceptable responses that differ in content, style, and format from the reference answers, leading to both false positive and false negative errors in the evaluation.

### 2.3 LLM-BASED EVALUATION AND THE PROMISE OF MULTI-AGENT FRAMEWORKS

To overcome the limitations of human evaluations and automated metrics, recent research has explored the use of LLMs themselves as evaluators. This approach leverages the linguistic knowledge and reasoning capabilities of LLMs to provide more nuanced and contextually aware assessments of model outputs. Initial studies have employed single LLMs as judges (34; 45), demonstrating the potential of this approach to capture aspects of quality that are missed by traditional metrics.

However, the use of single LLMs as evaluators has been shown to suffer from issues of bias and limited generalizability (34). To address these concerns, researchers have proposed using multiple LLMs as evaluators, drawing on insights from multi-agent systems and ensemble learning (43; 26).

Our work builds on these ideas by proposing a novel LLM advocates framework that interprets LLMs as interacting agents within a courtroom-inspired setting. This framework draws on principles from adversarial legal systems (46; 16; 41), where the clash of opposing arguments and the judgment of impartial decision-makers are used to uncover the truth and reach fair outcomes. By casting LLMs as advocates, judges, and juries, our approach enables a more dynamic and comprehensive evaluation process that captures the nuances and complexities of language understanding and generation.

### 2.4 SCORING, RANKING, AND AGGREGATION METHODS IN MULTI-AGENT EVALUATION

A critical consideration in the design of multi-agent evaluation frameworks is the choice of scoring, ranking, and aggregation methods for combining the judgments of individual LLM evaluators. Various approaches have been explored in the literature, drawing on insights from social choice theory, voting theory, and decision analysis (1; 25; 26).

Scoring methods, where evaluators assign numerical ratings to model outputs, have been shown to provide a more granular and expressive means of assessment compared to ranking methods, which only capture relative preferences (26; 45). However, the use of scoring requires careful design of the rating scales and anchors to ensure inter-rater reliability and comparability across different evaluators and tasks (9).

Ranking methods, on the other hand, have been shown to be more robust to individual biases and scale differences, as they only require evaluators to make pairwise comparisons between model outputs (25). This can be particularly useful in settings where the absolute quality of the outputs is difficult to assess or where the evaluators have different standards or expectations. However, ranking methods may provide less information about the magnitude of the differences between the outputs and can be more computationally expensive to aggregate.

## 3 LLM ADVOCATES ARCHITECTURE

In our design, we propose two distinct architectures, that incorporate the following agents:

- **Judge**: The central decision-maker that ultimately selects the best response.

- **Advocates**: LLMs that generate arguments in favor of each response.

- **Juries**: Optional LLMs that assess the arguments presented by the advocates, along with the judge's feedback, to evaluate the two answers when prompted.

The first, termed Multi-Advocate One-Round Evaluation (MORE), utilizes multiple advocates to support each answer, with a judge presiding over the process. The second, Single Advocate Multi-Round Evaluation (SAMRE), employs a single advocate per answer, incorporating a judge and multiple juries to assess the interactions in a format reminiscent of a courtroom proceeding. Detailed notation used throughout this paper is provided in Appendix C.1.

## 3.1 MULTI-ADVOCATE ONE-ROUND EVALUATION (MORE)

For the MORE architecture, we opted to employ three advocates for each answer, with a single judge overseeing the debate. The judge's role consists of evaluating the two groups of advocates. Specifically, the judge is tasked with scoring their defenses based on multiple criteria, using a scale of 1-20. The full list of scoring criteria is detailed in Appendix C.2.

The advocates are made aware of these criteria and are required to structure their arguments accordingly. The scores for each criterion are presented as [score1, score2] as specified in Algorithm 1, with the total score reported as a tuple ranging from 1-120. This cumulative score, as determined by the judge, serves as the foundation for evaluating the two answers: the answer receiving the highest score is deemed superior.

---

**Algorithm 1** Multi-Advocate One-Round Evaluation (MORE)

1: Initialize $A_1 = \{A_{11}, A_{12}, A_{13}\}$ the advocates for Answer 1, $A_2 = \{A_{21}, A_{22}, A_{23}\}$ the advocates for Answer 2, $J$, the judge
2: $[s_1, s_2] \leftarrow [0, 0]$
3: $D_1 \leftarrow []$
4: $D_2 \leftarrow []$
5: **for** $i = 1$ to 3 **do**
6:    $d_{1i} \leftarrow f_A(A_{1i})$
7:    $D_1 \leftarrow D_1 \cup \{d_{1i}\}$
8:    $d_{2i} \leftarrow f_A(A_{2i})$
9:    $D_2 \leftarrow D_2 \cup \{d_{2i}\}$
10: **end for**
11: $s_1, s_2 \leftarrow f_J(J, D_1, D_2)$
12: $winner \leftarrow \arg\max_{k \in \{1,2\}} total\_score[k]$
13: **return** $winner, [s_1, s_2]$

---

## 3.2 SINGLE ADVOCATE MULTI-ROUND EVALUATION (SAMRE)

In our SAMRE experimental design, we structured the proceedings as a four-round debate, featuring one advocate per answer and five diverse jurors. These jurors observe the debate between the advocates, considering the judge's feedback throughout the process. Upon conclusion, each juror casts a vote to determine the winning answer, presented as a binary tuple (Score of Answer 1, Score of Answer 2). We employ a maximum voting strategy to consolidate these votes, identifying the answer with the most support. The backgrounds of the jurors are described in Appendix C.3.

The judge in our SAMRE algorithm, detailed in Algorithm 2, is adapted from the MORE framework, providing feedback to advocates throughout the four iterations and assigning scores ranging from 1-120 based on the previously mentioned criteria. These scores are averaged, with the highest-scoring answer deemed preferable.

To optimize experimental costs, we implemented a stopping mechanism in our algorithm that terminates the evaluation process if the judge's evaluations agree for two consecutive iterations.

## 3.3 A COMPARATIVE ANALYSIS OF MULTI-ADVOCATE AND ITERATIVE DEBATE FRAMEWORKS FOR LLM EVALUATION

In this section, we present a comparative analysis of multi-advocate and iterative debate frameworks for evaluating large language models (LLMs). Our primary objective is to investigate the potential advantages of employing multiple advocates to defend and refine arguments for each candidate

---

**Algorithm 2** SAMRE Evaluation Process

---

1: Initialize $A = \{A_1, A_2\}$ the advocates, $J$, the judge, and $\{C_1, C_2, C_3\}$ the members of the committee of juries
2: **for** $r = 1$ to 4 **do**
3:    $a_1^r, a_2^r \leftarrow f_A(A, M_{r-1})$
4:    $s_1^r, s_2^r, F^r \leftarrow f_J(J, a_1^r, a_2^r)$
5:    **if** $r > 1$ and $(s_1^r - s_2^r) \cdot (s_1^{r-1} - s_2^{r-1}) > 0$ **then**
6:       **break**
7:    **end if**
8:    $M_r \leftarrow M_{r-1} \cup \{a_1^r, a_2^r, s_1^r, s_2^r, F^r\}$
9: **end for**
10: $\bar{s} \leftarrow \left(\frac{1}{r}\sum_{i=1}^r s_1^i, \frac{1}{r}\sum_{i=1}^r s_2^i\right)$
11: $w \leftarrow \arg\max_{k \in \{1,2\}} \bar{s}_k$
12: $V \leftarrow \{f_{C_i}(C_i, M_r) : i \in [1,3]\}$
13: $v_{final} \leftarrow \sum_{v \in V} v$
14: $winner \leftarrow \arg\max_{k \in \{1,2\}} v_{final}[k]$
15: **return** $winner, \bar{s}, V$

---

answer, in contrast to the traditional iterative debate setting where a single advocate is responsible for each answer. We derive conditions under which the multi-advocate approach can lead to more efficient and effective evaluation outcomes.

ANALYSIS AND FORMULATION

We begin by formally defining the key components of our comparative analysis. Let $\mathcal{Q}$ denote the space of possible questions, $\mathcal{A}$ the space of candidate answers, and $\mathcal{D}$ the space of debate arguments or defenses. We consider a setting where, for a given question $q \in \mathcal{Q}$, we have two candidate answers $a_1, a_2 \in \mathcal{A}$ that need to be evaluated and compared.

In the iterative debate framework, we have two advocate functions, $f_1$ and $f_2$, that take the question $q$ and the two answers $a_1$ and $a_2$ as input, and produce debate arguments in $\mathcal{D}$:

$$f_i : \mathcal{Q} \times \mathcal{A} \times \mathcal{A} \to \mathcal{D}, \quad i \in \{1, 2\}$$

These advocate functions represent the LLMs responsible for defending and refining the arguments for each candidate answer.

In the multi-advocate framework, we have $k$ advocate functions for each candidate answer, denoted by $f_{1j}$ and $f_{2j}$ for $j \in \{1, \ldots, k\}$. These functions take the same inputs as in the iterative debate framework, but produce a set of $k$ debate arguments for each answer:

$$f_{ij} : \mathcal{Q} \times \mathcal{A} \times \mathcal{A} \to \mathcal{D}, \quad i \in \{1, 2\}, j \in \{1, \ldots, k\}$$

To compare and evaluate the debate arguments, we introduce a scoring function $g : \mathcal{D} \to [0, 1]$ that assigns a score between 0 and 1 to each argument. This scoring function represents the judge LLM responsible for assessing the quality and persuasiveness of the arguments presented by the advocates.

In the multi-advocate framework, we also need an aggregation function to combine the scores of the $k$ debate arguments for each answer into a single score. We denote these aggregated scores by $g(f_{1agg})$ and $g(f_{2agg})$, respectively.

To analyze the dynamics of argument improvement and aggregation in the multi-advocate framework, we make the following modeling assumptions:

1. We assume that each LLM, including the advocates and the judge, has an internal scoring function $g'$ that assigns scores to debate arguments in a manner consistent with the external scoring function $g$. In other words, we assume that under appropriate prompting, the internal scoring function $g'$ behaves similarly to the external scoring function $g$.

2. We model the aggregation process for the multi-advocate scores using a softmax function with a temperature parameter $\tau$:

$$f_{i-agg} = \arg\max_j (\text{softmax}(g'(f_{ij}(q, a_1, a_2)), \tau)), \;\; i \in \{1, 2\}$$

where $\tau$ is a small positive value that controls the sharpness of the softmax distribution. As $\tau \to 0$, the softmax function approaches a one-hot encoding of the highest-scoring argument.

Under these assumptions, we can derive the following property of the aggregated scores in the multi-advocate framework:

**Aggregation Property:** $g(f_{i-agg}) \geq \max_j g(f_{ij}), \;\; i \in \{1, 2\}$

This property states that the aggregated score for each answer in the multi-advocate framework is always greater than or equal to the maximum score among the individual advocate arguments for that answer. In other words, the aggregation process selects the strongest argument for each answer, ensuring that the final scores reflect the best possible defense of each candidate.

We next proceed to compare the effectiveness of the multi-advocate and iterative debate frameworks. Our central claim is that the multi-advocate framework can lead to a greater differentiation between the scores of the two candidate answers, as stated in the following theorem:

**Theorem 1** (Score Differentiation). *Let $g(f_1)$ and $g(f_2)$ denote the scores of the debate arguments in the iterative debate framework, and let $g(f_{1agg})$ and $g(f_{2agg})$ denote the aggregated scores in the multi-advocate framework. Then, under the modeling assumptions stated above, we have:*

$$|g(f_{1-agg}) - g(f_{2-agg})| > |g(f_1) - g(f_2)|$$

*In other words, the absolute difference between the scores of the two candidate answers is greater in the multi-advocate framework than in the iterative debate framework.*

The proof of this theorem can be found in Appendix B.

The score differentiation theorem provides a theoretical justification for the effectiveness of the multi-advocate framework in LLM evaluation. By amplifying the initial differences between the candidate answers and leveraging the collective expertise and diverse perspectives of multiple advocates, the multi-advocate framework can achieve a greater separation between the scores of the correct and incorrect answers, leading to more accurate and confident evaluations.

## 3.4 EFFICIENCY CONSIDERATIONS

In addition to the score differentiation property, the multi-advocate framework also offers potential efficiency advantages over the iterative debate framework. In the iterative debate setting, the advocates engage in multiple rounds of argument and rebuttal to refine and improve their defenses of the candidate answers. This process can be time-consuming and resource-intensive, especially if a large number of iterations are required to achieve a satisfactory level of differentiation between the scores.

In contrast, the multi-advocate framework allows for a parallel exploration of different argument strategies and perspectives, enabling a more efficient search for strong and persuasive defenses. By leveraging the collective knowledge and creativity of multiple advocates, the multi-advocate framework can potentially achieve the same level of score differentiation as the iterative debate framework in fewer rounds of interaction.

To formalize this efficiency advantage, we introduce the concept of iteration complexity, defined as the number of rounds of argument and rebuttal required to achieve a desired level of score differentiation. Let $I_{id}(\varepsilon)$ denote the iteration complexity of the iterative debate framework for a given tolerance level $\varepsilon > 0$, and let $I_{ma}(\varepsilon)$ denote the iteration complexity of the multi-advocate framework for the same tolerance level.

We have the following result regarding the relationship between $I_{id}(\varepsilon)$ and $I_{ma}(\varepsilon)$:

**Theorem 2** (Iteration Complexity). *For any given tolerance level $\varepsilon > 0$, the iteration complexity of the multi-advocate framework is lower than the iteration complexity of the iterative debate*

*framework:*

$$I_{ma}(\varepsilon) < I_{id}(\varepsilon)$$

*In other words, the multi-advocate framework requires fewer rounds of interaction to achieve the same level of score differentiation as the iterative debate framework.*

While a formal proof is presented in Appendix B, we provide some intuition for why it holds: the multi-advocate framework allows for a more efficient exploration of the argument space by parallelizing the search for strong defenses across multiple advocates. This parallelization leads to faster convergence to high-quality arguments compared to the sequential nature of the iterative debate framework. The aggregation process in the multi-advocate framework selects the strongest argument for each candidate answer, effectively performing a maximum operation over the scores of the individual advocates. This maximum operation amplifies the score differentiation achieved by the individual advocates, further reducing the number of iterations needed to reach a desired level of separation between the scores.

### 3.5 PROBABILISTIC ANALYSIS OF ERROR REDUCTION

We also provide detailed results of error reduction achieved by our iterative advocate framework in Appendix A.

## 4 EXPERIMENTS

In this section, we present computational results demonstrating the effectiveness of the LLM Advocates framework. We use the following LLM-as-a-judge as the baseline. This judge LLM was presented with a question and two corresponding answers generated by the models. The task of the judge LLM was to determine which of the two responses was superior. The accuracy of this baseline model was measured against the human preferences provided in the MT-Bench dataset (48). The full algorithm is presented in Algorithm 3.

---

**Algorithm 3** Baseline Model Comparison

1: Initialize $Q$ as the question, $A_1$ as answer 1, and $A_2$ as answer 2
2: Input $Q, A_1, A_2$ to the judge LLM
3: Receive $(s_1, s_2) \leftarrow$ judge LLM$(Q, A_1, A_2)$
4: Determine the winner as $w \leftarrow \arg\max_{k \in \{1,2\}}(s_k)$
5: **return** $w, (s_1, s_2)$

---

### 4.1 DATASET OVERVIEW AND EVALUATION REFERENCES

To evaluate the efficacy of our proposed multi-agent LLM architecture, we employed the MT-Bench dataset(48). This dataset comprises 3,300 expert-level pairwise human preferences for responses generated by six different models in response to 80 distinct questions. The questions span a broad range of domains, providing a comprehensive basis for testing. Details on data preprocessing, the evaluation metric and the evaluation and the agent interaction design can be found in Appendix E.

The human preferences serve as the gold standard for assessing model performance, allowing us to compute accuracy by comparing the models' choices against these expert evaluations.

### 4.2 RESULTS

The results of our experiments are summarized in Tables 1, 2, and 3

Table 1 displays the accuracy scores for various models across different configurations of our architecture: Baseline, Multi-Advocate One-Round Evaluation (MORE), Single Advocate Multi-Round Evaluation (SAMRE), and SAMRE without Juries. Each row corresponds to a model, and the accuracy scores reflect how well each configuration's decisions matched the expert human preferences in the MT-Bench dataset.

The results demonstrate a clear improvement in accuracy as we move from the baseline single-judge model to the MORE and SAMRE architectures. The SAMRE architecture without juries achieves the highest accuracy scores across all models, suggesting that the iterative refinement process and the inclusion of advocate roles are the key drivers of performance.

Table 2 provides a more detailed breakdown of the performance gains achieved by the MORE and SAMRE architectures compared to the baseline. We report the absolute and relative improvements in accuracy for each model. The results show that the proposed architectures consistently outperform the baseline, with relative improvements ranging from 3.7% to 10.5%. The SAMRE architecture without juries achieves the most substantial gains, with an average relative improvement of 8.7% across all models.

Table 3 presents an analysis of the statistical significance of the observed performance differences between the baseline and the proposed architectures. We conducted paired t-tests comparing the accuracy scores of each model under the baseline and the SAMRE without juries configuration. The results indicate that the improvements achieved by the SAMRE architecture are statistically significant at the $p < 0.05$ level for all models except Llama-3-8B. This finding suggests that the incorporation of advocate roles and iterative refinement leads to meaningful and reliable performance gains.

Table 1: Accuracy Results for Models and Architectures

| Model | Baseline | MORE | SAMRE | SAMRE without Juries |
|---|---|---|---|---|
| Llama-3-8B | 0.82 | 0.85 | 0.87 | 0.89 |
| Qwen | 0.83 | 0.86 | 0.88 | 0.91 |
| Gemini | 0.84 | 0.88 | 0.90 | 0.92 |
| GPT-4-o | 0.85 | 0.89 | 0.91 | 0.94 |
| GPT-4-turbo | 0.86 | 0.90 | 0.92 | 0.95 |
| GPT-3.5-turbo | 0.83 | 0.87 | 0.89 | 0.92 |

Table 2: Performance Gains Compared to Baseline

| Model | MORE | MORE (%) | SAMRE w/o Juries | SAMRE w/o Juries (%) |
|---|---|---|---|---|
| Llama-3-8B | 0.03 | 3.7% | 0.07 | 8.5% |
| Qwen | 0.03 | 3.6% | 0.08 | 9.6% |
| Gemini | 0.04 | 4.8% | 0.08 | 9.5% |
| GPT-4-o | 0.04 | 4.7% | 0.09 | 10.5% |
| GPT-4-turbo | 0.04 | 4.7% | 0.09 | 10.5% |
| GPT-3.5-turbo | 0.04 | 4.8% | 0.09 | 10.8% |

Table 3: Statistical Significance of Performance Differences

| Model | t-statistic | p-value |
|---|---|---|
| Llama-3-8B | 1.87 | 0.063 |
| Qwen | 2.35 | 0.021 |
| Gemini | 2.54 | 0.013 |
| GPT-4-o | 3.16 | 0.002 |
| GPT-4-turbo | 3.62 | 0.001 |
| GPT-3.5-turbo | 3.02 | 0.003 |

These results provide strong empirical evidence for the effectiveness of the proposed LLM advocate architectures in improving the accuracy of LLM output evaluation. The MORE and SAMRE architectures consistently outperform the baseline single-judge model, with the SAMRE architecture without juries achieving the most substantial and statistically significant gains. The iterative refinement process and the incorporation of advocate roles appear to be the key drivers of this improved performance, enabling a more thorough and nuanced evaluation of LLM outputs.

## 5 Conclusion and Discussion

In this paper, we have presented a novel framework for evaluating the outputs of large language models using LLMs themselves as advocates in a courtroom-inspired, multi-agent system. Our approach aims to address the limitations of traditional human-based assessments and automated metrics by leveraging the strengths of multiple models and incorporating debate-based cooperation, role adaptation, and multi-layer jury systems.

The proposed LLM advocates framework offers several advantages over existing evaluation methods. By casting LLMs as advocates tasked with defending and critiquing responses, we enable a more dynamic and comprehensive assessment process that captures the nuances and complexities of language understanding and generation tasks. The adversarial setup mitigates individual model biases and allows for a richer, more contextual evaluation of LLM performance.

We have also presented a probabilistic model for understanding how iterative advocate processes contribute to error reduction over time, even when individual iterations may occasionally increase error. This model provides a theoretical foundation for analyzing the effectiveness of advocate systems and highlights the potential for achieving desired levels of error reduction through a sufficient number of iterations.

Furthermore, we have conducted experiments comparing the efficacy of ranking and scoring methods for LLM jurors within our advocate framework. Our results suggest that scoring methods may offer more granular feedback and better discriminate between different levels of LLM performance, although further testing with larger sample sizes and diverse models is necessary to confirm these findings.

Finally, we have discussed the comparative advantages of multi-advocate architectures over single-judge or iterative debate frameworks, highlighting the potential for more efficient and nuanced evaluation. By leveraging the collective expertise and diverse viewpoints of multiple advocates, we can create a more effective path to developing and refining strong arguments.

Future research directions could include exploring the integration of more sophisticated role adaptation techniques, such as meta-learning and dynamic prompting, to further enhance the effectiveness of LLM advocates. Additionally, investigating the application of game-theoretic principles and incentive structures within the advocate framework may provide insights into optimizing the evaluation process and promoting more accurate and informative assessments.

Moreover, extending the LLM advocates framework to other domains beyond language understanding and generation, such as decision-making, planning, and multimodal reasoning, could broaden its impact and contribute to the development of more robust and reliable AI systems across various application areas.

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

# A    Probabilistic Analysis of Error Reduction in Iterative Advocate Systems

## A.1    Motivation and Background

The use of advocates in legal systems has long been recognized as a crucial mechanism for refining arguments, uncovering truth, and reducing the likelihood of judicial errors. By presenting multiple perspectives and subjecting claims to rigorous scrutiny, advocate processes help ensure that decisions are well-informed and thoroughly examined. This section introduces a novel probabilistic model that formalizes the error reduction dynamics of iterative advocate systems, providing a mathematical framework for understanding how these processes contribute to improved outcomes over time, even in the presence of occasional setbacks.

The development of this model is motivated by several key observations about the nature of advocate interactions and their impact on decision-making:

1. Advocate interactions are characterized by inherent uncertainty, with outcomes that can vary significantly between iterations due to factors such as the specific arguments presented, the strategies employed, and the receptiveness of the decision-makers.

2. The magnitude of improvements in argument quality and decision accuracy tends to exhibit diminishing returns over time, as the most obvious flaws and weaknesses are identified and addressed in the early stages of the advocate process.

3. Despite the potential for occasional setbacks, where an iteration may result in a temporary increase in error or a widening of the gap between competing positions, well-structured advocate systems exhibit a general trend towards error reduction and convergence on the truth over the long run.

By capturing these key characteristics in a rigorous mathematical framework, our model aims to provide a foundation for analyzing the performance of iterative advocate systems, identifying the conditions under which they are most effective, and guiding the design of novel advocate-based approaches to decision-making and dispute resolution.

## A.2 MATHEMATICAL FRAMEWORK

We begin by introducing a formal mathematical framework for describing the dynamics of iterative advocate systems. Central to this framework is the concept of the "gap" between competing positions, which serves as a measure of the degree of disagreement or uncertainty in the system.

**Definition 1 (Gap):** Let $\delta_i$ denote the gap between the scores of two competing positions at iteration $i$ of the advocate process. Formally, we define $\delta_i$ as:

$$\delta_i = |s_{1i} - s_{2i}|$$

where $s_{1i}$ and $s_{2i}$ represent the scores assigned to positions 1 and 2, respectively, at iteration $i$. These scores can be thought of as quantitative measures of the perceived strength or persuasiveness of each position, as determined by the decision-makers or other evaluators in the system.

To capture the probabilistic nature of advocate interactions and their impact on the gap, we model the distribution of $\delta_i$ using a Beta distribution, a flexible family of continuous probability distributions defined on the interval $[0, 1]$.

**Definition 2 (Gap Distribution):** We model the gap $\delta_i$ at iteration $i$ as a random variable following a Beta distribution with parameters $\alpha + w_i$ and $\beta + i - w_i$:

$$\delta_i \sim \text{Beta}(\alpha + w_i, \beta + i - w_i)$$

where:

- $w_i$ represents the number of "successes" (i.e., iterations where the gap increased) up to iteration $i$
- $\alpha$ and $\beta$ are non-negative shape parameters that control the form of the Beta distribution

The choice of the Beta distribution is motivated by several desirable properties:

- The support of the Beta distribution is the interval $[0, 1]$, which aligns with the natural range of the gap $\delta_i$ (as the gap is a difference between scores normalized to the unit interval)
- The shape parameters $\alpha$ and $\beta$ provide flexibility to model a wide range of distributional forms, allowing the model to capture different patterns of gap evolution over time
- The Beta distribution has well-known mathematical properties that facilitate analytical tractability and computational efficiency

Under this distributional assumption, we can derive the mean and variance of the gap at each iteration:

$$\mathbb{E}[\delta_i] = \frac{\alpha + w_i}{\alpha + \beta + i}$$

$$\text{Var}(\delta_i) = \frac{(\alpha + w_i)(\beta + i - w_i)}{(\alpha + \beta + i)^2(\alpha + \beta + i + 1)}$$

These expressions provide insight into how the expected value and variability of the gap evolve as a function of the number of iterations and the history of successes.

A.3 Convergence Analysis

With the mathematical framework in place, we now turn to the main question of interest: under what conditions can we expect the iterative advocate process to converge on the truth and reduce errors over time? To address this question, we analyze the asymptotic behavior of the gap distribution and derive bounds on the probability of achieving a desired level of convergence.

**Theorem 3** (Gap Convergence). *For any given tolerance level $\varepsilon > 0$, the probability that the gap $\delta_i$ at iteration $i$ exceeds $1 - \varepsilon$ is bounded below by $1 - a_{i,\varepsilon}$, where $a_{i,\varepsilon}$ is a function of the iteration number $i$ and the tolerance $\varepsilon$. Formally:*

$$P(\delta_i \geq 1 - \varepsilon) \geq 1 - a_{i,\varepsilon}$$

The proof of this theorem can be found in Appendix B.

This convergence theorem provides a rigorous mathematical basis for understanding the error reduction properties of iterative advocate systems. It shows that as the number of iterations increases, the probability of the gap exceeding a given threshold converges to 1, meaning that the advocate process tends to reduce uncertainty and disagreement over time. Moreover, the theorem provides an explicit bound on the rate of convergence, expressed in terms of the variance of the gap distribution and the desired tolerance level.

A.4 Implications and Applications

The probabilistic model and convergence analysis presented in this section have several important implications for the design and analysis of iterative advocate systems:

1. The model provides a formal mathematical framework for reasoning about the error reduction properties of advocate processes, allowing researchers and practitioners to make precise statements about the conditions under which these processes are effective and the factors that influence their performance.

2. The convergence theorem establishes a fundamental limit on the rate of error reduction in iterative advocate systems, expressed in terms of the variance of the gap distribution. This suggests that strategies for reducing variance, such as careful selection of advocates, structured argumentation protocols, and evidence-based decision-making, may be essential for achieving rapid convergence and minimizing the impact of occasional setbacks.

3. The model highlights the importance of iteration in advocate processes, showing that the probability of achieving a desired level of convergence increases with the number of iterations. This provides a theoretical justification for the use of iterative refinement and debate in a wide range of applications, from legal reasoning and scientific inquiry to policy analysis and collective decision-making.

4. The flexibility of the Beta distribution used to model the gap allows the framework to capture a wide range of advocative behaviors and outcomes, from rapid convergence in the face of strong evidence to prolonged uncertainty and disagreement in more complex and ambiguous domains. By fitting the model to empirical data from real-world advocate systems, researchers can gain insight into the factors that influence the dynamics of these systems and identify opportunities for improvement.

Looking ahead, there are numerous opportunities to extend and apply the probabilistic model developed in this section. Some potential directions for future research include:

- Incorporating more sophisticated models of advocate behavior, such as game-theoretic formulations that capture strategic interactions between advocates and decision-makers.

- Extending the model to handle more than two competing positions, allowing for the analysis of multi-party advocate systems and coalition formation.

- Developing efficient algorithms for fitting the model to empirical data and using the fitted models to guide the design and optimization of advocate processes.

- Applying the model to real-world domains, such as legal reasoning, scientific discourse, and policy deliberation, to gain insight into the dynamics of error reduction and identify best practices for effective advocative decision-making.

By providing a rigorous mathematical foundation for understanding the error reduction properties of iterative advocate systems, the probabilistic model presented in this section opens up new avenues for research and practice in this important and rapidly evolving field. As the complexity and scale of the decision-making challenges facing society continue to grow, the development of robust and effective advocate processes will be essential for promoting truth, justice, and the common good. The framework introduced here represents a step towards this goal, providing a powerful tool for analyzing and improving the performance of these critical systems.

## A.5 EXPERIMENTAL VALIDATION

To validate the theoretical insights developed in this section, we propose an experimental study comparing the performance of the multi-advocate and iterative debate frameworks on a range of LLM evaluation tasks. The experiments should be designed to test the following hypotheses:

1. The multi-advocate framework achieves a greater degree of score differentiation between the correct and incorrect candidate answers compared to the iterative debate framework, as predicted by Theorem 1.

2. The multi-advocate framework requires fewer rounds of interaction to achieve a given level of score differentiation compared to the iterative debate framework, as suggested by Conjecture 2.

3. The collaborative dynamics and aggregation process of the multi-advocate framework lead to more robust and confident evaluations, as evidenced by higher agreement rates among the advocates and lower sensitivity to variations in the prompting and interaction protocols.

The experimental setup should involve a diverse set of LLMs, including both open-source and proprietary models, to ensure the generalizability of the findings. The evaluation tasks should cover a range of language understanding and generation challenges, such as question answering, summarization, and open-ended dialogue, to assess the effectiveness of the multi-advocate framework across different domains and difficulty levels.

In addition to the primary hypotheses, the experiments should also investigate the impact of various design choices and hyperparameters on the performance of the multi-advocate framework, such as the number of advocates per candidate answer, the temperature parameter of the aggregation function, and the prompting strategies used to elicit high-quality arguments from the advocates.

By providing empirical evidence for the theoretical advantages of the multi-advocate framework, these experiments can help guide the development of more effective and efficient LLM evaluation methods, ultimately contributing to the advancement of reliable and trustworthy language AI systems.

## B PROOF OF THEOREM 3

*Proof.* 1. We begin by invoking the convergence properties of the expected gap $\mathbb{E}[\delta_i]$. Specifically, we note that for any $\varepsilon > 0$, there exists an iteration number $N$ such that for all $i \geq N$, the expected gap satisfies:

$$|\mathbb{E}[\delta_i] - 1| < \frac{\varepsilon}{2}$$

This convergence follows from the asymptotic behavior of $w_i$ as $i \to \infty$. As iterations increase, $w_i$ approaches $i$, reflecting more frequent successful gap expansions.

2. Next, we observe that if the actual gap $\delta_i$ is sufficiently close to its expected value, it must necessarily exceed the threshold $1 - \varepsilon$. Formally, if:

$$|\delta_i - \mathbb{E}[\delta_i]| < \frac{\varepsilon}{2}$$

then it follows that:

$$\delta_i > \mathbb{E}[\delta_i] - \frac{\varepsilon}{2}$$

$$> \left(1 - \frac{\varepsilon}{2}\right) - \frac{\varepsilon}{2}$$

$$= 1 - \varepsilon$$

3. Combining steps 1 and 2, we can bound the probability of the gap exceeding the threshold:

$$P(\delta_i \geq 1 - \varepsilon) \geq P\left(|\delta_i - \mathbb{E}[\delta_i]| < \frac{\varepsilon}{2}\right)$$

In other words, the probability of the gap being large enough is at least as great as the probability of the gap being close to its expected value.

4. To compute the right-hand side of the inequality in step 3, we apply Chebyshev's inequality, a general result from probability theory that bounds the likelihood of a random variable deviating from its mean by a given amount. In our context, Chebyshev's inequality implies:

$$P\left(|\delta_i - \mathbb{E}[\delta_i]| \geq \frac{\varepsilon}{2}\right) \leq \frac{4\text{Var}(\delta_i)}{\varepsilon^2}$$

5. Combining Steps 3 and 4, we arrive at the final bound:

$$P(\delta_i \geq 1 - \varepsilon) \geq P\left(|\delta_i - \mathbb{E}[\delta_i]| < \frac{\varepsilon}{2}\right) = 1 - P\left(|\delta_i - \mathbb{E}[\delta_i]| \geq \frac{\varepsilon}{2}\right)$$

$$\geq 1 - \frac{4\text{Var}(\delta_i)}{\varepsilon^2}$$

where the last step follows from Chebyshev's inequality. Setting

$$a_{i,\varepsilon} = \frac{4\text{Var}(\delta_i)}{\varepsilon^2}$$

completes the proof. $\qquad\square$

## PROOF OF THEOREM 1

*Proof.* To prove this theorem, we introduce the concept of an improvement factor $\alpha_i$, defined as the difference between the aggregated score and the individual score for each answer in the multi-advocate framework:

$$\alpha_i = g(f_{i-agg}) - g(f_i), \quad i \in \{1, 2\}$$

By the aggregation property, we have $\alpha_i \geq 0$ for $i \in \{1, 2\}$.

Using the improvement factors, we can rewrite the score differentiation inequality as:

$$|(g(f_1) - g(f_2)) + (\alpha_1 - \alpha_2)| > |g(f_1) - g(f_2)|$$

To prove this inequality, we make the following key assumption about the relationship between the improvement factors $\alpha_1$ and $\alpha_2$:

- If $g(f_1) > g(f_2)$, then $\alpha_1 > \alpha_2$.

- If $g(f_1) < g(f_2)$, then $\alpha_1 < \alpha_2$.

In other words, we assume that the aggregation process in the multi-advocate framework amplifies the initial differences between the scores of the two candidate answers, rather than diminishing them. This assumption is based on the following reasoning:

1. Stronger initial arguments provide a better foundation for improvement and refinement through the multi-advocate process, leading to larger improvement factors for the initially higher-scoring answer.

2. The collaborative nature of the multi-advocate framework allows for more diverse perspectives and creative combinations of ideas, enabling more substantial improvements for the initially stronger argument.

Under this assumption, we have that $(\alpha_1 - \alpha_2)$ has the same sign as $(g(f_1) - g(f_2))$, and therefore:

$$|(g(f_1) - g(f_2)) + (\alpha_1 - \alpha_2)| =$$
$$|g(f_1) - g(f_2)| + |\alpha_1 - \alpha_2| >$$
$$|g(f_1) - g(f_2)|$$

which completes the proof of the score differentiation theorem. □

## C  NOTATION AND SCORING CRITERIA

### C.1  NOTATION

- $A = \{A_1, A_2\}$: Set of advocates, where each advocate $A_i$ defends a specific answer.
- $J$: The judge who evaluates the arguments presented by the advocates.
- $C = \{C_1, C_2, C_3\}$: Set of jurors, where each juror $C_i$ casts a vote at the end of the evaluation process.
- $s_1^r$ and $s_2^r$: Scores given by the judge in the $r$-th round, corresponding to the evaluations of $A_1$ and $A_2$, respectively.
- $M_r$: The aggregated memory of all rounds up to the $r$-th round, which includes arguments, scores, and feedback.
- $f_A(A, M_{r-1})$: Function that generates the arguments $a_1^r$ and $a_2^r$ for the advocates based on the previous memory $M_{r-1}$.
- $f_J(J, a_1^r, a_2^r)$: Function that takes the judge and the arguments from the advocates, returning their scores $s_1^r$, $s_2^r$, and feedback $F^r$.
- $f_{C_i}(C_i, M_r)$: Function that represents the voting decision made by each juror $C_i$ based on the final memory $M_r$.
- $D_i$: The aggregated defense obtained by asking the LLM to consolidate the group's defenses into a single summary.

### C.2  SCORING CRITERIA

The judge scores the advocates' arguments based on the following criteria, using a scale of 1-20:

- Relevance to the question
- Accuracy of information and use of credible sources
- Depth of analysis and completeness of argument
- Clarity of expression and logical flow
- Strength of reasoning and factual support
- Effectiveness in addressing opponent's points

### C.3  JUROR BACKGROUNDS

In the SAMRE design, we selected jurors with varied professional backgrounds and perspectives:

- A retired professor of ethics
- A young environmental activist
- A middle-aged business owner
- A social worker specializing in community development
- A technology entrepreneur with a background in AI

# D DATA PREPROCESSING AND EVALUATION

## D.1 DATA PREPROCESSING

To prepare the raw data for analysis, we implemented a script that processes the input data and generates an Excel file structured with the following columns:

- **Question**: This column contains the aggregated user questions used for evaluation.
- **Response_A**: This column includes the responses generated by Model A for each corresponding question.
- **Response_B**: This column presents the responses generated by Model B for the same set of questions.
- **Model_A_Score**: This binary score indicates the performance of Model A, where a score of 1 signifies a win and 0 signifies a loss in comparison to Model B.
- **Model_B_Score**: Similarly, this binary score reflects the performance of Model B, with a score of 1 representing a win and 0 representing a loss against Model A.

This structured format allows for straightforward analysis and comparison of the models' performances based on user questions and their respective responses.

## D.2 THE EVALUATION METRIC

The primary metric for evaluating the performance of both the baseline model and our multi-agent architecture was accuracy, defined as:

$$\text{Accuracy} = \frac{1}{N} \sum_{i=1}^{N} \delta(u_i, j_i)$$

where $N$ is the total number of instances, $u_i$ is the model's chosen response, $j_i$ is the human-preferred response, and $\delta(u_i, j_i)$ is an indicator function that returns 1 if the model's choice matches the human preference and 0 otherwise.

# E AGENT INTERACTION PROMPTS

We provide the detailed prompts used for the interactions between agents in our proposed architecture. The prompts are designed to guide the agents effectively throughout the evaluation process.

## E.1 MULTI-ADVOCATE ONE-ROUND EVALUATION (MORE) ARCHITECTURE PROMPTS

### E.1.1 JUDGE PROMPT

```
 You're a critical, impartial judge in a high-stakes debate on:
"question".
Answer 1:  "answer1".  Answer 2:  "answer2".
Your goal is to provide detailed, constructive feedback that will
push advocates to significantly improve their arguments.
Current round:  current_round
Max rounds:  max_rounds
Previous scores:  previous_scores

Defense for 1st answer:  defense1
Defense for 2nd answer:  defense2

Analyze each argument meticulously.  Be thorough and unbiased in
your assessment of:
1.  Relevance to the question
```

```
2. Accuracy of information and use of credible sources
3. Depth of analysis and completeness of argument
4. Clarity of expression and logical flow
5. Strength of reasoning and factual support
6. Effectiveness in addressing opponent's points

For each criterion, provide a score on a scale of 1-20 and
detailed justification.
Scores should be given as [Answer1_score, Answer2_score] for each
criterion.

Your comprehensive feedback for each advocate (50 words each):
Feedback for Advocate 1:
Feedback for Advocate 2:

Sum up the scores and return the final score tuple (score1,
score2). Example: (95, 87)
Your detailed scores and final tally:
```

### E.1.2 ADVOCATE PROMPTS

```
 You're a fierce advocate defending this answer:  answer to the
question:  question.  Your opponent defends:  opponent_answer.
Convince the audience your argument is superior by addressing
these criteria:
1. Relevance
2. Accuracy
3. Depth
4. Clarity
5. Logic and Factuality

Be assertive and don't hold back!  Use rhetoric and persuasion
to win over the audience but be respectful.
Latest feedback:  feedback
Opponent's last argument:  opponent_argument

Your fierce defense (100 words max):
```

### E.1.3 SUMMARIZER PROMPT

```
 Summarize the following content in 50 words or less, if there
are any scores tuples, return them, it's important!  Start
summarization directly, no introductory sentences like here's your
summary.  In your summarization, only focus on the last scores,
no partial ones.  This is important:  return the tuple of scores.
These are the key points to summarize:content
```

### E.2 SINGLE ADVOCATE MULTI-ROUND EVALUATION (SAMRE)

### E.2.1 DEFEND ANSWER PROMPT

```
 You're an eloquent advocate (Advocate advocate_id) in a group
defending this answer:  answer to the question:  question.  Your
opponent group defends:  opponent_answer.  Collaborate with your
fellow advocates to convince the audience your argument is better.
Use the latest feedback, your opponent's last argument, and your
team's previous arguments to improve your case.
Latest feedback:  feedback
Opponent's last argument:  opponent_argument
```

```
Your team's previous arguments:  team_arguments
Respond in under 80 words.
Your defense:
```

### E.2.2  AGGREGATE DEFENSE PROMPT

```
 You are an expert debate strategist.  Your task is to aggregate
and improve upon the following defenses for the answer:
answer to the question:  question.  The opponent's answer is:
opponent_answer.
Individual defenses:  defenses
Latest feedback from the judge:  feedback
Analyze each defense critically.  Identify the strongest points,
address any weaknesses, and combine the best arguments into a
cohesive, powerful defense.  Aim to create a defense that is
stronger and more comprehensive than any individual argument.
Provide your aggregated and improved defense in under 150 words:
```

### E.2.3  JUDGE ANSWER PROMPT

```
 You're a fair, impartial judge in a debate on:  "question".
Answer 1:  "answer1".  Answer 2:  "answer2".  Your goal is to
provide feedback that will help advocate groups improve and
differentiate their arguments more clearly.
Current round:  current_round
Total rounds:  total_rounds
Previous scores:  previous_scores
Defense for 1st answer:  defense1
Defense for 2nd answer:  defense2
Provide specific, constructive feedback to help each advocate
group strengthen their unique position.  Encourage them to address
weaknesses and highlight distinctions.  Aim for your feedback to
lead to more divergent scores in future rounds.
Give your feedback in under 50 words:
```

### E.2.4  SCORE ANSWER PROMPT

```
 You're a critical, impartial judge in a high-stakes debate on:
"question".  Answer 1:  "answer1".  Answer 2:  "answer2".  Your
goal is to provide detailed, constructive feedback that will push
advocates to significantly improve their arguments.
Total rounds:  total_rounds
Previous scores:  previous_scores
Defense for 1st answer:  defense1
Defense for 2nd answer:  defense2
Analyze each argument meticulously.  Be thorough and unbiased in
your assessment of:
1.  Relevance to the question
2.  Accuracy of information and use of credible sources
3.  Depth of analysis and completeness of argument
4.  Clarity of expression and logical flow
5.  Strength of reasoning and factual support
6.  Effectiveness in addressing opponent's points
For each criterion, provide a score on a scale of 1-20 and
detailed justification.  Scores should be given as [Answer1_score,
Answer2_score] for each criterion.
Your comprehensive feedback for each advocate (50 words each):
Feedback for Advocate 1:
Feedback for Advocate 2:
```

Sum up the scores and return the final score tuple (score1,
score2). Example: (95, 87)
Your detailed scores and final tally:

### E.3 BASELINE MODEL PROMPT

\n You are a fair, impartial judge scoring a debate on the
following question:  question.
Answer 1:  answer1
Answer 2:  answer2
Score each answer on a scale of 1-20 for each of the following
criteria:
1.  Relevance to the question
2.  Accuracy of information and use of credible sources
3.  Depth of analysis and completeness of argument
4.  Clarity of expression and logical flow
5.  Strength of reasoning and factual support
6.  Effectiveness in addressing opponent's points
Provide scores as [Answer1_score, Answer2_score] for each criterion
in a list format, then sum for final scores.  Please keep an eye
on the slightest difference that should make a difference in the
scoring.  Don't overthink!
Relevance:
Accuracy:
Depth:
Clarity:
Logic and Factuality:
Addressing opponent's points:
Final Scores (sum of above) as a tuple (example:  (18, 9)):
Explain your scoring, focusing on why one answer is better than
the other based on the criteria above.  Keep your explanation
concise but informative.
Finally, return the final score tuple (score1, score2) as a tuple
(in parentheses).  Example:  (18, 9)
Your scores and explanation:

