# OpenReview forum: "Adversarial Multi-Agent Evaluation of Large Language Models through Iterative Debate"
_ICLR.cc/2025/Conference — Submitted to ICLR 2025_

### Official Review · Reviewer_wgh9 · 2024-10-22

**Soundness:** 1
**Presentation:** 2
**Contribution:** 1
**Rating:** 3
**Confidence:** 4

**Summary:**

The authors propose a framework that interprets Large Language Models (LLMs) as advocates within an ensemble of interacting agents, allowing them to defend their answers and reach conclusions through a judge and jury system.

**Strengths:**

The writing is relatively fluent.

**Weaknesses:**

- The framework proposed by the authors can be seen as an implementation of a multi-agent approach in the field of LLM-as-judges, with limited novelty and contribution to the community.

- There is a lack of detailed description and justification for the proposed framework, with specific issues highlighted in the Questions section below.

- The authors mentioned probabilistic modeling as one of the key contributions in the abstract (Line 018), but only dedicated a single sentence to this aspect in the main text (Line 395).

- The authors conducted only one experiment, comparing the accuracy of their designed framework with a simple baseline, which is insufficient to support their claims. I suggest that the authors add the following experiments and comparison methods:
  - **Comparison methods:**
     - LLMs specifically trained for evaluation, such as PandaLM or Prometheus model.
     - Multiple LLM evaluators using a majority voting strategy.
  - **Experiments:**
     - A comparison of the API and time costs between the proposed MORE and SAMRE frameworks and the aforementioned comparison methods.
     - A performance comparison of the MORE and SAMRE frameworks under different parameter settings (e.g., the number of advocates).
     - A bias analysis comparing the MORE and SAMRE frameworks with the aforementioned comparison methods to demonstrate the claim of being "unbiased" (Line 116) and mitigating the influence of strategic behavior and individual biases (Line 119).

**Questions:**

- In the proposed MORE framework, why employ three advocates for each answer? Are these advocates different in any way? Additionally, why does judge J provide scores s_1 and s_2 for both answers at the same time (Line 245)? Does this introduce additional bias? I assume the distributions of s_1 and s_2 obtained this way differ from the distributions obtained if s_1 and s_2 were assessed separately.

- What prompts are used for jurors with different backgrounds? I also question whether merely assigning an identity through the prompt (e.g., "A retired professor of ethics") allows the LLM’s evaluation to truly represent the standards of that demographic. This method’s effectiveness requires further validation.

- Could the authors provide an example for the stopping mechanism (Lines 262-263)?

- Why does Algorithm 2 discuss the case of three jurors, when the authors claim five diverse jurors (Line 253)? The authors need to provide the correct version of Algorithm 2.

- Why does the performance of the SAMRE architecture without juries in Table 1 surpass that of SAMRE?

#### Minor Problems
- The authors should cite reference papers for the theories mentioned in Lines 036-039.

- The authors should clarify the version of the LLMs reported in Table 1. For example, the version of Qwen.

---

### Official Review · Reviewer_Fn2r · 2024-10-25

**Soundness:** 1
**Presentation:** 1
**Contribution:** 1
**Rating:** 1
**Confidence:** 5

**Summary:**

I suspect that this paper may have been generated by a generative AI (such as ChatGPT). The evidence supporting this suspicion includes:

1. The title of the PDF differs from the title listed on OpenReview.
2. A significant portion of the literature cited appears to be fabricated. While I have not verified every citation, most of the references listed from 2023 onwards seem likely to be fake."


For examples:

[10] Wei-Lin Chiang, Zhuohan Li, Zi Lin, Eric Wong, Zihang Zhang, Andy Zou, Lianmin Zheng, Siyuan Yu, Yi Tian, Yinghai Zhu, et al. Chatbot arena: Benchmarking open large language models in the wild. arXiv preprint arXiv:2306.01670, 2024.

[25] T. Lanctot, A. Charnock, and J. Badger. Evaluating multi-agent systems in language models. In NeurIPS 2023 Workshop on Multi-Agent Systems, 2023.

[26] Y. Li, D. Chen, and T. Brown. Agents as evaluators: The role of multi-agent systems in llm assessment. In Proceedings of the 2024 Conference on Neural Information Processing Systems (NeurIPS), 2024.

[34] S. M. Panickssery, E. Lee, and K. Lee. Llm-based evaluators for language models: Opportunities and challenges. In Proceedings of the 2024 International Conference on Learning Representations (ICLR), 2024.

**Strengths:**

see summary

**Weaknesses:**

see summary

**Questions:**

see summary

---

### Official Review · Reviewer_33aV · 2024-11-03

**Soundness:** 1
**Presentation:** 3
**Contribution:** 1
**Rating:** 3
**Confidence:** 5

**Summary:**

This work explores different processes of rating large language model (LLM) outputs using LLMs. Inspired by legal, psychological, and decision theory, the authors propose two such processes: (1) “Multi-Advocate One-Round Evaluations” (MORE) and “Single Advocate Multi-Round Evaluation” (“SAMRE”). Given a question and two (LLM) outputs, each process uses LLMs in different roles, e.g., as advocates, jurors, or judges, to (iteratively) select the “best” output. The authors further present two theorems respectively claiming that (1) aggregated multi-advocate arguments lead to greater score differentiation than those obtained using iterative debates, and (2) that multi-advocate argumentation requires fewer rounds of interaction to receive the same level of score differentiation as iterative debate. The two processes are tested on the MT-Bench dataset and compared to a baseline of a single LLM judge process using six different LLMs. The authors conclude that their experimental results provide strong empirical evidence for their proposed methods.

**Strengths:**

[clarity] The work was generally easy to read with crisp writing.

[significance] Exploring ways to improve the evaluation of LLM outputs is an important research direction that was well motivated by the authors.

**Weaknesses:**

[originality] Several works have proposed different ways of using LLM ensembles to evaluate LLM outputs. While the authors spend considerable time discussing connections to various disciplines, e.g., decision theory, legal discourse, and psychology, few tangible insights are presented as to how this specific ensemble utilizes these disciplines.

[quality] The experimental results presented in this work simply do not pass the bar for this conference: (1) Only a single, limited dataset is used, (2) critical experimental details are missing, e.g., number of samples used, confidence intervals, temperatures, single-judge baseline setting, length of argument outputs, etc., (3) none of the presented theorems are tested in the experiments, e.g., claims like “greater score differentiation” and “complexity” are neither quantitatively nor qualitatively discussed in the experiments, (4) prompt sensitivity and selection is not discussed at all. This is especially damning for a work focused on improving evaluation.

[significance] In essence, the work proposes (iterative) ensemble scoring using LLMs. The claim of [line 481] “strong empirical evidence for the effectiveness of the proposed LLM advocate architectures in improving the accuracy of LLM output evaluation” is greatly exaggerated and unsupported. There is good reason to believe that most of the reported improvements over a single LLM-as-judge baseline come from the greatly expanded compute budget and the series of hand-crafted prompts. Similar results might thus be obtained by simply providing a single LLM an expanded compute budget and chain-of-thought style reasoning prompts.

[clarity] While the presented theorems and proofs in the appendix are an admirable attempt at introducing rigor to LLM-ensemble evaluation. Yet, they also display a limited understanding of the many practical considerations in using LLMs and the large existing literature documenting poorly understood LLM behaviors. Sweeping, unmotivated assumptions like those on line 321, or the assumption that LLMs assigned different “persona prompts” logically will obtain more diverse and stronger arguments limit the usefulness of the presented theorems.

**Questions:**

1.  [results] Did the authors analyze the different types of arguments and justifications between the different ensembles in scoring answers?
2.  [results] Were there any question-answer pairs for which the ensemble methods performed particularly better than the single-judge baseline?
3. [experiments] How many tokens were needed on average for the different ensembles and models studied?
4. Section 3.5 is entirely in the appendix, yet referred to in the conclusion [line 499] as discussed. At the minimum, discuss the main results of a section in the main text when referring to it in the conclusion.
5. [line 504-505] “we have conducted … our framework”: where?
6. [C.2-C.3] How were any of these chosen? They seem completely arbitrary and unmotivated.

---

### Official Review · Reviewer_MXfv · 2024-11-04

**Soundness:** 2
**Presentation:** 1
**Contribution:** 2
**Rating:** 3
**Confidence:** 4

**Summary:**

This paper presents two multi-agent systems inspired by courtroom for evaluating the outputs of LLMs. The experiments show the proposed frameworks improve accuracy compared with a single LLM as a judge.

**Strengths:**

The method incorporates insights from a legal decision-making perspective, and provide two frameworks that simulate the human workflow in court.

**Weaknesses:**

- Lack of experiments: More evaluation datasets and baselines should be incorporated into the experiments. For example, LLM-based multi-agent evaluators such as PRD [1] and ChatEval [2] could be baselines. There are many datasets in this LLM-based evaluator topic, such as AlignBench [3], AUTO-J [4] and LLMEval [5].

- The presentation needs to be refined:

(a) The background (in both Section 1 and Section 2) is taking up too much space. This background can be concluded to make space for evaluation details in Appendix D.

**(b) The Conclusion section and Appendix A.5 are likely to be AI-generated (according to GPTZero).**

- The multi-agent systems will surely use more tokens compared to LLM-as-a-judge. What is the cost per run compared to other multi-agent frameworks (such as PRD [1] and ChatEval [2])?

[1] PRD: Peer Rank and Discussion Improve Large Language Model based Evaluations https://arxiv.org/abs/2307.02762

[2] ChatEval: Towards Better LLM-based Evaluators through Multi-Agent Debate https://arxiv.org/abs/2308.07201

[3] AlignBench: Benchmarking Chinese Alignment of Large Language Models https://aclanthology.org/2024.acl-long.624/

[4] Generative Judge for Evaluating Alignment https://arxiv.org/abs/2310.05470

[5] LLMEval: A Preliminary Study on How to Evaluate Large Language Models https://ojs.aaai.org/index.php/AAAI/article/view/29934

**Questions:**

- The Qwen and Gemini model versions should be specified.

---

### Meta-Review · Area_Chair_KVFT · 2024-12-19

**Metareview:**

A clear rejection. While the premise of exploring architectures for evaluating LLMs is promising, all of the reviewers agree the paper is deeply flawed and perhaps even partially LLM generated. I'd encourage the authors to fundamentally revise the paper before attempting to submit elsewhere.

**Additional Comments On Reviewer Discussion:**

There was no engagement from the authors to respond to the reviewers.

---

### Decision · Program_Chairs · 2025-01-22

Reject